# Improving Building Extraction by Using Knowledge Distillation to Reduce the Impact of Label Noise

**Gang Xu** [1,2], **Min Deng** [2], **Geng Sun** [2], **Ya Guo** [2] **and Jie Chen** [2,*]

1   School of Artificial Intelligence, Zhejiang College of Security Technology, Wenzhou 325016, China
2   School of Geosciences and Info-Physics, Central South University, Changsha 410083, China
*   Correspondence: cj2011@csu.edu.cn

**Abstract:** Building extraction using deep learning techniques has advantages but relies on a large number of clean labeled samples to train the model. Complex appearance and tilt shots often cause many offsets between building labels and true locations, and these noises have a considerable impact on building extraction. This paper proposes a new knowledge distillation-based building extraction method to reduce the impact of noise on the model and maintain the generalization of the model. The method can maximize the generalizable knowledge of large-scale noisy samples and the accurate supervision of small-scale clean samples. The proposed method comprises two similar teacher and student networks, where the teacher network is trained by large-scale noisy samples and the student network is trained by small-scale clean samples and guided by the knowledge of the teacher network. Experimental results show that the student network can not only alleviate the influence of noise labels but also obtain the capability of building extraction without incorrect labels in the teacher network and improve the performance of building extraction.

**Keywords:** building extraction; knowledge distillation; label noise

## 1. Introduction

Remote sensing image building extraction plays an important role in urban planning [1], disaster assessment [2], land resource management [3], and smart city construction [4]. With the continuous development of Unmanned Aerial Vehicle (UAV) platforms and sensor technologies, several images with high spatial resolution are increasingly accessible and provide an important data source for building extraction [5].

The high resolution images can provide rich spectral and textural information of buildings [6], which has markedly contributed to the progress of automatic building extraction research. However, the task of building extraction from high-resolution remote sensing images still presents considerable challenges due to the variable and diverse appearance of buildings [7]. Specifically, the varying size and complex appearance of buildings cause large intra-class variation in buildings and poor generalization of extraction algorithms [8]. A large similarity is also found between the backgrounds of buildings and non-buildings, and the background information substantially interferes with building extraction. Moreover, due to the existence of skewed sensor shooting and sun altitude angles, buildings in the images often have different degrees of tilt, geometric deformation, and appear to be obscured by shadows or obstacles, with high-rise buildings becoming the most evident, adding difficulty to the challenge of building extraction [9].

In recent years, with the development of deep learning and artificial intelligence technologies, convolutional neural networks (CNNs) have played a remarkable role in high-resolution remote sensing image building extraction with their excellent feature extraction and representation capabilities [10–16]. Compared with the traditional methods using low-level features [17–19], the performance of CNNs has been remarkably improved [20]. Most of the building extraction methods using CNNs are full convolutional networks

(FCNs) [21] for binary semantic segmentation. In FCN, the fully connected layer of CNN is replaced by a standard convolutional layer to implement an end-to-end deep learning architecture for semantic segmentation. This model has an encoder–decoder structure [22]; that is, the resolution of the feature map is reduced by a downsampling operation in the encoder, and the decoder gradually recovers the resolution of the feature map and obtains the prediction results. This end-to-end learning approach allows for modeling using a combination of hierarchical features and contextual information to predict buildings from their context accurately [20]. With the proposed transformer architecture, researchers have recently found that processing images using a transformer can overcome the inability of FCN to model global contextual information [23]. Therefore, using a transformer instead of FCN for feature extraction has turned out to be a mainstream direction in computer vision tasks [24]. Models such as Swin Transformer [25] and SegFormer [26] have been successfully applied to the semantic segmentation task of buildings and achieved better results than FCN.

Deep learning methods rely on large-scale well-labeled training samples; once the number of samples is insufficient or the sample labels of a certain category contain errors, the prediction results of the model will be seriously affected. The labeling of building samples is usually obtained by manual visual interpretation; if the high-rise buildings in the images are severely tilted or obscured by obstacles, such as shadows, then manual labeling will become increasingly difficult and result in labeling errors. In addition, the annotated samples can be derived from ground survey data, but updating these samples in real time is difficult due to the high update cost. Therefore, guaranteeing the current nature of ground survey data are difficult, and mislabeling is inevitable when used with current images (As shown in Figure 1a, the difference in shooting time and angle resulted in the vector data not matching the roof of the building.). Most of the current building extraction models are variants of FCNs proposed to address the aforementioned building extraction challenges [10–16]. However, these methods are only effective when the labeled samples are correct; once trained with samples containing incorrect labels, the resulting models fail to perform effective building extraction [27].

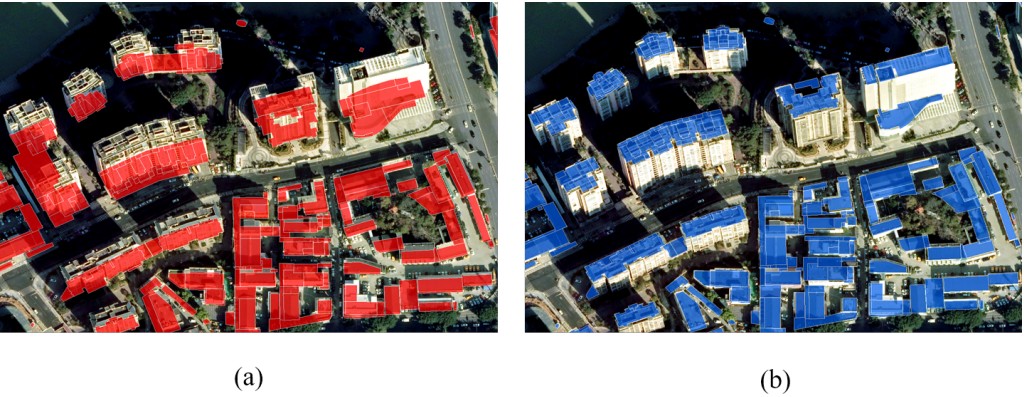

　　　　　　　　　　(a)　　　　　　　　　　　　　　　　　　　　　　　　　　(b)

**Figure 1.** (**a**) Noise-containing building labels, where the red label does not match the roof of the building, are marked on the side of the building, especially in high-rise buildings; (**b**) corrected building label, where the blue label matches the building roof perfectly.

The problem of noisy building labels can be alleviated by retraining with label-tolerant learning methods if the training data are available in its entirety again. The error tolerant learning method can retrain the model using a small number of clean labeled samples that are manually corrected and all of the noisy labeled samples, allowing the model to be guided by the clean labeled samples and be able to tolerate category labeling errors. For example, Mnih and Hinton [28] constructed a simple labeled noise model for a deep neural network-based patch classifier to extract buildings from aerial images. A remarkably complex noise model representing the relationship between images, noise labels, real labels,

and noise types was proposed in [29] for image classification. Inspired by the label noise probability model, an additional layer, placed on top of the last softmax layer of the original network, is proposed in [30,31] to capture the relationship between clean and noisy labels, but with a different optimization strategy. However, if the training data are only partially available again, then the fault-tolerant learning approach cannot retrain the model if the training set is missing.

The mislabeling in this portion of the training data that can be obtained again must be corrected to mitigate the mislabeling effect of the building sample. Training a new model directly on the corrected data can solve the problem of noisy labels, but the small size of the training sample often leads to model overfitting [32]. Therefore, a combination of models trained with noisy labels and this small portion of corrected training data are needed to minimize the effect of noisy labels while maintaining the generalizability of the model across building types obtained with a large number of trained samples. Fine-tuning the noise-labeled trained model using corrected small training samples is generally a common practice inspired by transfer learning theory. For example, Maggiori et al. [20] first pre-trained the FCN using large noise-labeled data generated by GIS and then fine-tuned it using a small amount of manually labeled clean data. Fine-tuning guides the model with the correct sample labels, which reduces the effect of label noise, but also allows the model to lose the capability to classify buildings without labeling error types to some extent. Therefore, developing a model that mitigates the effects of label noise while maintaining the capability of the model is necessary to extract buildings without label noise types.

Overall, to address the problem of insufficient generalization of the model for the extraction of different types of buildings when using a small number of clean samples, this paper proposes a practical approach to obtain knowledge of each building type from a model trained with a large number of noisy building samples by using an advanced transformer-based model and the concept of knowledge distillation [33], while also reducing the effect of noisy labels. The framework comprises two similar teacher and student networks, where the feature extractors are Swin Transformer [25] and the decoders are both UperNet [34], which are designed to perform high-resolution remote sensing image building extraction but using different inputs. The teacher network is trained by large-scale noise-labeled samples. The extraction of buildings with noisy label types is poor. However, the teacher network can have better extraction results than the student network on building types without noise due to the large volume of data. The student network is trained by small-scale clean-labeled samples. While the student network is trained, the teacher network provides guidance and permits students to access the knowledge in the teacher network, allowing the student network to mitigate the effects of noisy labels while gaining the capability to extract buildings without mislabeled categories in the teacher network.

The paper is organized as follows: Section 1 introduces the research status of building extraction and the problems to be solved in this paper. Section 2 introduces the study area and data and explains the methodology, including details of the Swin Transformer-based semantic segmentation model for buildings and the knowledge distillation architecture. Section 3 presents the results. Section 4 discusses the results, and Section 5 concludes this paper.

## 2. Materials and Methods

### 2.1. Study Area and Data

Wenzhou City in Zhejiang Province, China was selected in this study as the study area. This area has a structurally and functionally rich building stock, as shown in Figure 2. Wenzhou is an important economic and cultural center along the southeast coast of China. Owing to its unique location, developed economy, and long history, Wenzhou has a complex and diverse building form, including high-rise commercial buildings, well-arranged residential houses, and low-rise flat buildings. Overall, the differences in building geometry, roof materials, and heights in the region introduce considerable challenges to the intelligent extraction of building roofs.

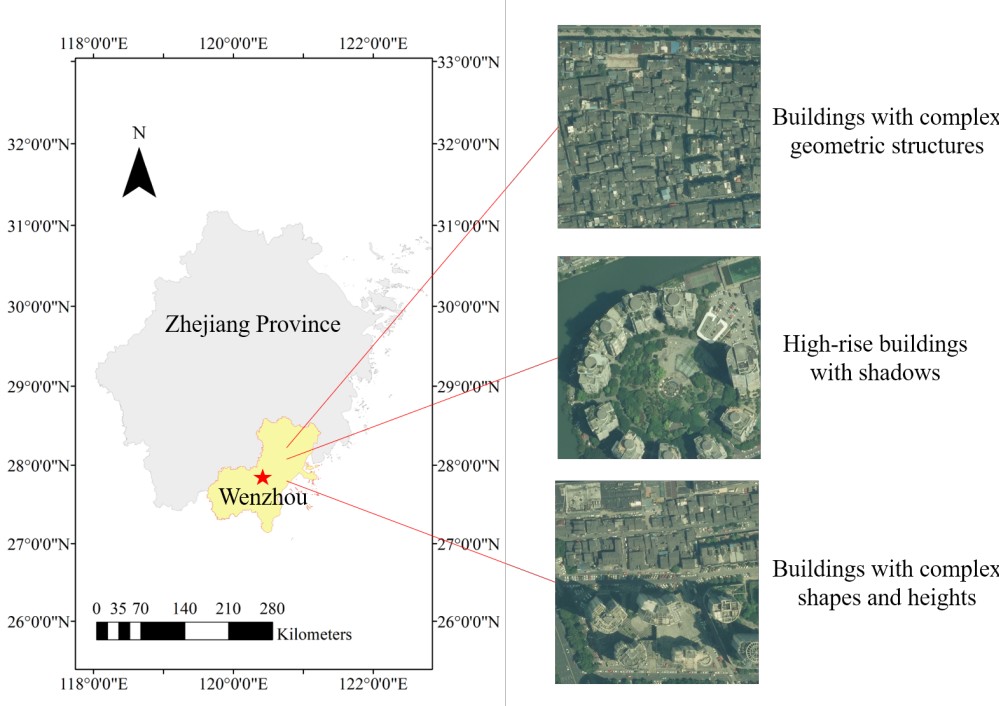

**Figure 2.** Location and diversity of buildings in the study area.

The data used to extract buildings in Wenzhou are aerial remote sensing images, including red, green, and blue bands (RGB), with a spatial resolution of 0.2 m. Two datasets containing annotations are used in this work: the large dataset (Dn) with noise and the small pure dataset (Dc). The difference between the two datasets is reflected in the images and annotations: the images of Dn and Dc are from the aerial remote sensing images of Wenzhou in 2021. The building labels of Dn are derived from the vector layer of buildings in Wenzhou, 2020, while those of Dc are manually outlined by human experts in aerial remote sensing images. All the remote sensing images and the vector data of Dn are from Tianditu Wenzhou [35].

Dn is the large dataset with noise, buildings in Dn are characterized by large data scales and complex structures. However, the building roof labels of aerial remote sensing images are noisy mainly due to the following two reasons. First, the building labels come from the vector data of buildings measured in the field, which is the actual area occupied by buildings in the two-dimensional plane, rather than the building roofs in the aerial images. Second, the aerial imagery was imaged in 2021 while the vector data were produced in 2020; this temporal disparity also resulted in the presence of noise in the building roof labels.

The labels of the Dc small dataset were obtained from the visual interpretation of aerial remote sensing images by human experts. Only a dataset with an area of 70.78 km$^2$ was obtained due to data availability and labeling cost considerations. The Dc small dataset is located in the central area of Wenzhou city and contains many high-rise buildings with shadows.

The Dc small dataset is divided into training and test data, which are both distributed in different regions, to ensure the accuracy and validity of the experiments. The aerial remote sensing images and labels in the two regions were cropped into image patches of size 1024 × 1024, and the paired samples are shown in Figure 3. The final number of images used for training is 1500, and the number of images used for testing is 1482. Notably, the buildings in the test area are not only geometrically diverse but also have a considerable variance in height (3–300 m).

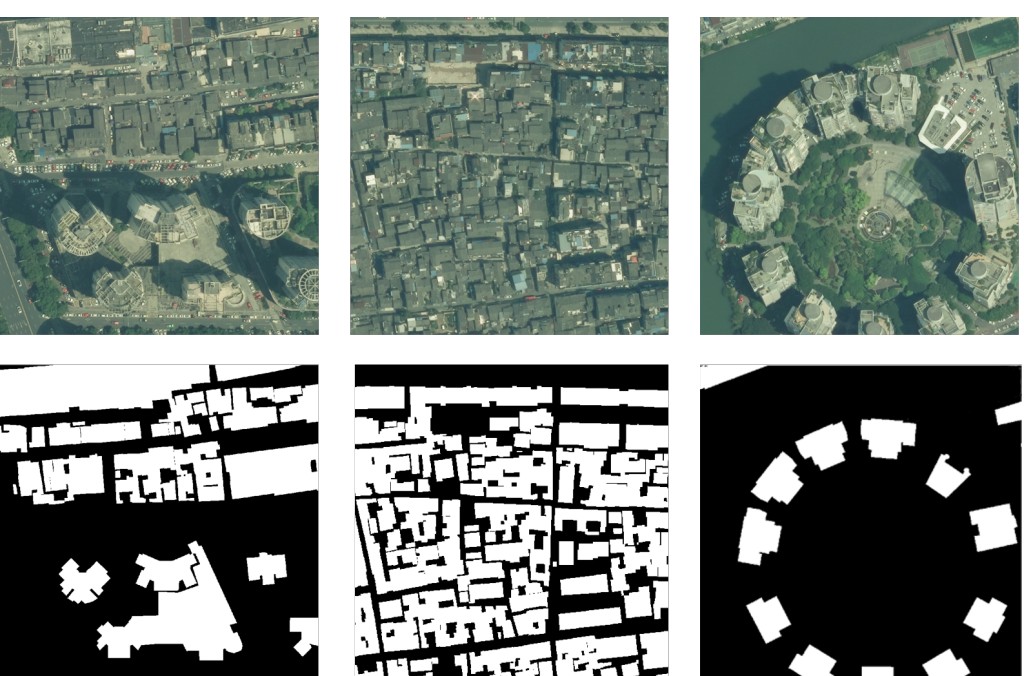

**Figure 3.** Pairs of images with labels. The top row is the aerial remote sensing image, and the bottom row is the corresponding building label.

*2.2. Methods*

2.2.1. Overall Architecture

The overall structure is shown in Figure 4, which comprises two sub-networks with identical structures, called the teacher and the student networks. Among them, the teacher network is trained from a large dataset of Dn containing noise; once the teacher network is trained to convergence, its model parameters will be frozen. This paper aims to take full advantage of the capability of the teacher network to generalize building features with structural diversity, combined with the potential advantage of a pure small data set for locating building areas, to train a student network with strong generalization for building roof extraction. The student network is trained and tested on the Dc small dataset and eventually used for the prediction of buildings throughout the study area.

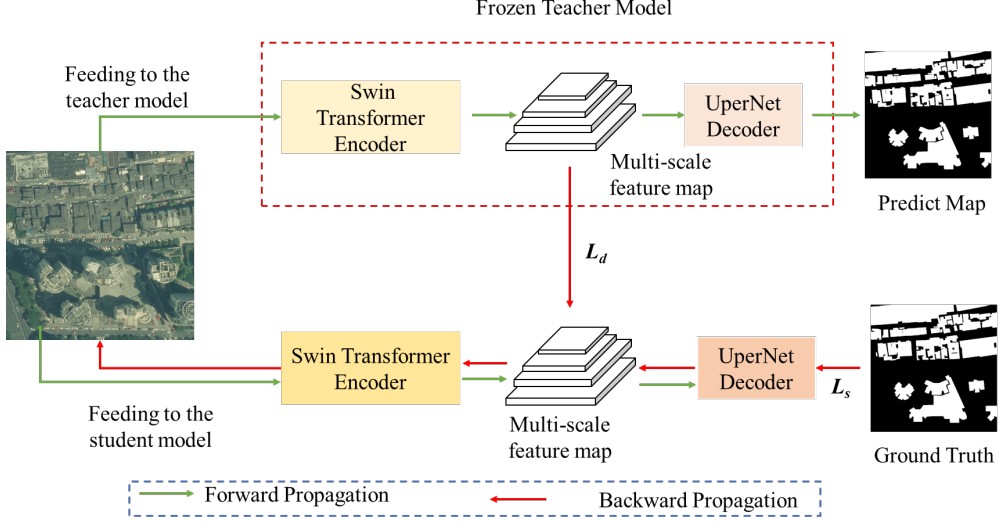

**Figure 4.** General structure diagram. During the training process, the parameters of the teacher model are frozen, and only the parameters of the student model are updated. Only the student model parameters must be used in the prediction process.

The teacher and student sub-networks contain a Swin Transformer feature encoder and an UPerNet feature decoder, respectively. The parameters of the teacher network are not updated, while the input of the student model includes aerial remote sensing images of Dc, and the output is a prediction map of buildings. In the training process of the student network, the aerial images of Dc are inputted to the teacher network with frozen parameters and the student network, and the data flow is shown by the green arrow in Figure 4. The gradient calculation of the student network parameters during backpropagation comprises two parts, as shown by the red arrow in Figure 4. That is, the loss function of the student network comprises two components, namely segmentation and distillation loss. Segmentation loss is the discrepancy between the building scores predicted by the student network and the real building labels, while distillation loss is the discrepancy between the multiscale feature maps extracted by the student and teacher networks. The former motivates the student model to locate and segment the roof area of the building accurately, and the latter forces the student model to be robust in extracting buildings with complex structures.

Swin Transformer is used as an encoder for image features considering the superior performance of the transformer family of structures on recent vision tasks to extract the rich local and global contextual information of buildings, and its structure is shown in Section 2.2.2 In addition, UPerNet is introduced as a decoder architecture considering the scale variability of buildings on images, and the structure is shown in Section 2.2.3. Finally, to increase the robustness of the student model and facilitate knowledge transfer from the teacher network to the student network, a distillation loss is introduced in the loss function to prompt the students to mimic the robust feature representation of the teacher network on the large-scale building dataset.

### 2.2.2. Swin Transformer Encoder

ViT is a transformer network for image classification tasks proposed by Google Brain in 2020. ViT uses pure transformers for the first time on vision tasks and achieves results comparable to state-of-the-art CNN networks. The main result of this work is the use of a transformer instead of a CNN and the demonstration that CNNs are not necessary for vision tasks. More specifically, ViT divides the input image into multiple patches ($16 * 16$ pixels), projects each patch into a fixed-length vector to feed the Transformer, and then repeatedly exploits the Transformer's global self-attention capability to capture the remote dependencies of the image. ViT achieves state-of-the art results on a number of benchmarks based on pre-training with large-scale datasets and uses fewer computational resources for training.

However, the direct use of ViT for intensive prediction tasks, such as target detection and semantic segmentation, would lead to the problem of large computational effort, that is, the computational complexity of ViT is quadratic to the size of the input image. Thus, the Swin Transformer uses the strategy of moving windows to compute the self-attention of local windows and the self-attention of cross-windows to address the aforementioned problem. This strategy not only reduces the computational complexity but also enhances the feature extraction capability of images. Inspired by this finding, this paper used the Swin Transformer as a feature encoder to extract the global and local information of buildings effectively, as shown in Figure 5.

The basic unit of the Swin Transformer is the Swin Transformer block, and its core is the W-MSA and SW-MSA modules. Two Swin Transformer blocks connected in series are shown in Figure 6. The Swin Transformer block includes the (Shifted) window-based self-attentive ((S)W-MSA) module, multilayer perceptron (MLP module), Layer Norm (LN) layer, and residual connection.

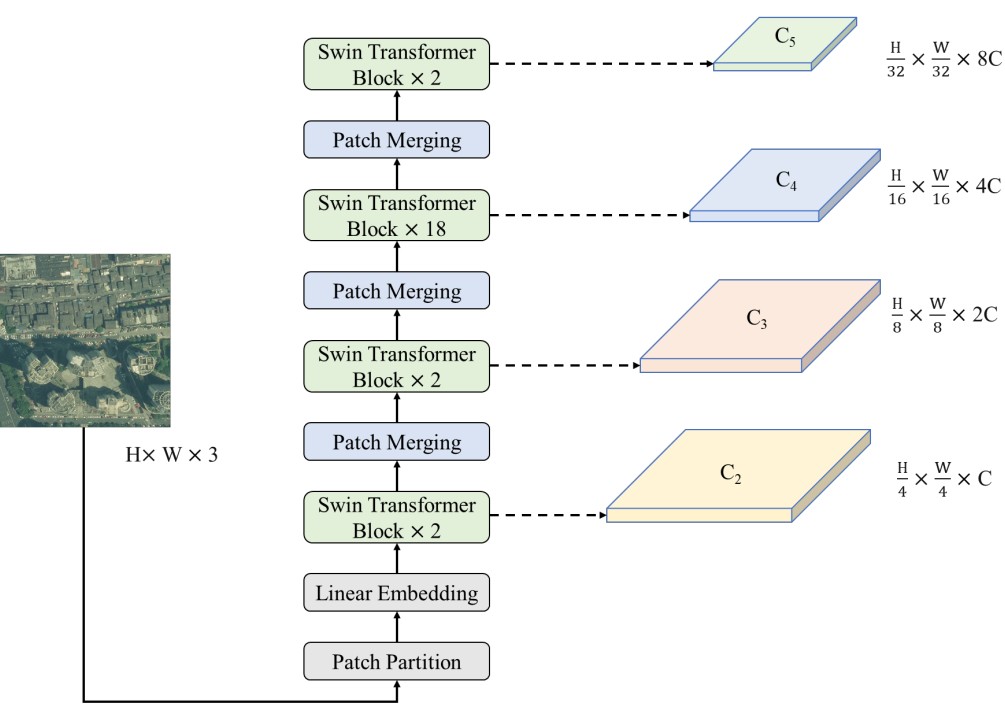

**Figure 5.** Swin Transformer Encoder. The input is satellite images of size H × W × 3, and the output is feature maps of four different scales.

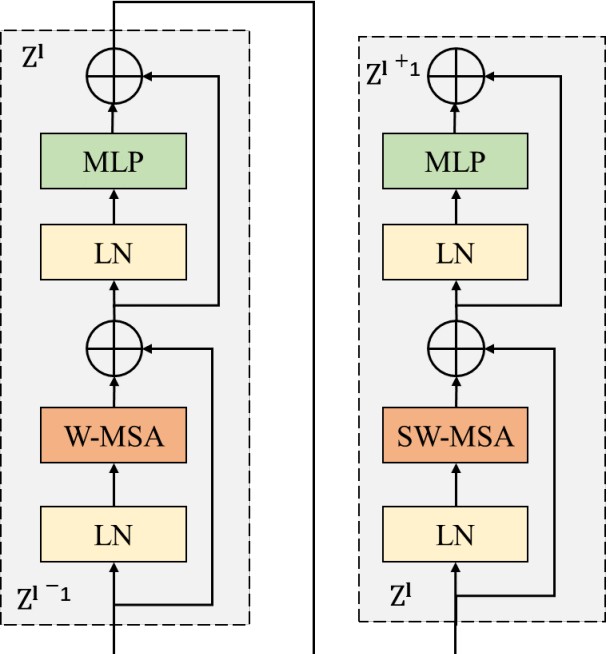

**Figure 6.** Two successive Swin Transformer blocks.

(a)   W-MSA module

ViT used standard multihead self-attention to compute global self-attention for the entire feature map, and its computational complexity limits its use in intensive prediction tasks. Swin Transformer proposes the W-MSA module to solve the problem. This transformer divides the feature map into multiple non-overlapping windows and computes the local self-attention in each window separately. The computational complexity is linearly related to the input image size.

As shown in Figure 7a, the W-MSA module divides the feature map into four $M \times M$ size windows and calculates the self-attention in each window separately. The window

self-attention in the W-MSA module is the same as the standard multiheaded self-attention, and the attention is calculated as follows:

$$Q = TW^q, \tag{1}$$

$$K = TW^k, \tag{2}$$

$$V = TW^v, \tag{3}$$

$$\text{Attention}(Q, K, V) = \text{Softmax}\left(\frac{QK^T}{\sqrt{m}}\right)V, \tag{4}$$

where $T \in \mathbb{R}^{M^2 \times d}$ denotes the self-attention input of self-attention, $W^q, W^k, W^v \in \mathbb{R}^{d \times d}$ denote the weights of the three linear mapping layers, and $Q, K, V \in \mathbb{R}^{M^2 \times d}$ denote query, key, and value, respectively. $d$ is a scaling factor to avoid the disappearance of the gradient of the Softmax function. $M^2$ denotes the spatial resolution, and $m$ denotes the channel dimension.

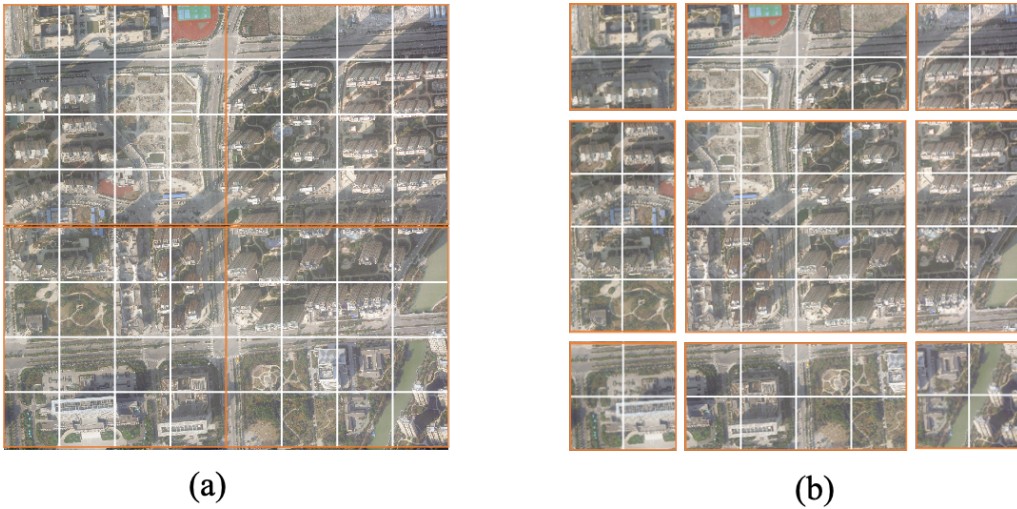

(a)            (b)

**Figure 7.** (**a**) Window partition strategy of W-MSA module; (**b**) window partition strategy of the SW-MSA module.

(b)    SW-MSA module

W-MSA performed the computation of attention only in the local window at a time and could not interact with the neighboring windows for information. The SW-MSA module was introduced to solve this problem. The window partitioning strategy of the SW-MSA module is shown in Figure 7b. Compared with the window division of W-MSA, the window of the SW-MSA module is shifted $(M/2, M/2)$. In the Swin Transformer, the blocks of the W-MSA module are alternated with those of the SW-MSA module to reduce the localization effect of the windowing strategy, as shown in Figure 6. The two consecutive Swin Transformers are calculated as follows:

$$\hat{z}^l = \text{WMSA}\left(\text{LN}\left(z^{l-1}\right)\right) + z^{l-1}, \tag{5}$$

$$z^l = \text{MLP}\left(\text{LN}\left(\hat{z}^l\right)\right) + \hat{z}^l, \tag{6}$$

$$\hat{z}^{l+1} = \text{SWMSA}\left(LN\left(z^l\right)\right) + z^l, \tag{7}$$

$$z^{l+1} = \text{MLP}\left(LN\left(\hat{z}^{l+1}\right)\right) + \hat{z}^{l+1}, \tag{8}$$

where $\hat{z}^l$ and $z^l$ denote the output of the *l*th Swin Transformer block (S)WMSA and MLP modules, respectively. The MLP comprises two fully connected layers with GELU nonlinear activation, and the LN layer is used before each (S)W-MSA module and each MLP.

(c)    Encoder

The encoder comprising Swin Transformer Block is shown in Figure 5, the input RGB image size is I $\in \mathbb{R}^{H \times W \times 3}$, and the size of each patch is set to $4 \times 4$. First, the patch partition module cuts the image into $\left(\frac{H}{4}\right) \times \left(\frac{W}{4}\right)$ tokens, and each token is flattened into 1D. The size of each token is $4 \times 4 \times 3 = 48$. Each token is then mapped to the specified dimension C using the linear embedding model, and the size of the feature map is $\left(\frac{H}{4}\right) \times \left(\frac{W}{4}\right) \times C$.

After the linear embedding layer, the feature map is passed through the Swin Transformer pyramid, which comprises multiple Swin Transformer blocks for global information interaction and patch merging modules for downsampling. After each patch merging in the encoding phase, the spatial resolution is reduced by a factor of 1 and the channel dimension becomes a factor of 2. After three stages, the size of the feature map becomes $\left(\frac{H}{32}\right) \times \left(\frac{W}{32}\right) \times 8C$. After the Swin Transformer encoder, four semantic levels of feature maps are obtained, namely $\left(\frac{H}{4}\right) \times \left(\frac{W}{4}\right) \times C$, $\left(\frac{H}{8}\right) \times \left(\frac{W}{8}\right) \times 2C$, $\left(\frac{H}{16}\right) \times \left(\frac{W}{16}\right) \times 16C$ and $\left(\frac{H}{32}\right) \times \left(\frac{W}{32}\right) \times 8C$, which will be denoted as $\{C_2, C_3, C_4, C_5\}$, respectively, and the downsampling multipliers are $\{4, 8, 16, 32\}$.

### 2.2.3. UPerNet Decoder

As shown in Figure 8, the student network of MSKD-BuildingNet used the multi-scale feature maps $\{C_2, C_3, C_4, C_5\}$ of the Swin Transformer encoder as the input of the UperNet decoder, and produced the prediction results of the buildings. The UPerNet decoder architecture is inspired by the feature pyramid network (FPN) and the PPM in PSPNet, which is shown in Figure 9.

The buildings in the test area remarkably vary in scale. Thus, aggregating contextual information from different regions is necessary to expand the perceptual field and improve the characterization of the buildings. Therefore, the pyramid pooling module (PPM) is added to the last layer of the Swin Transformer encoder. Specifically, different scales of pooling are used on the original feature map to obtain feature maps of different sizes, and then multiple sizes of feature maps containing the original feature map are stitched together in the channel dimension to finally output a conformed feature map incorporating multiple scales. In this work, the PPM module contains four layers, and the sizes of each layer are $1 \times 1$, $2 \times 2$, $3 \times 3$, and $6 \times 6$. The input of the PPM module is the feature map $C_5$, and the output is noted as $P_5$.

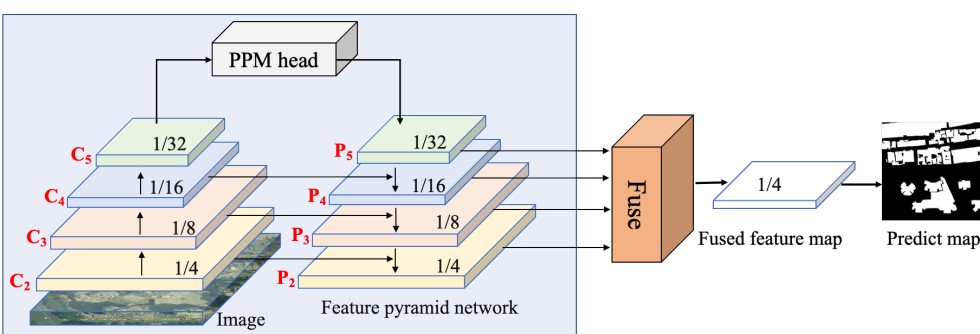

**Figure 8.** UPerNet decoder.

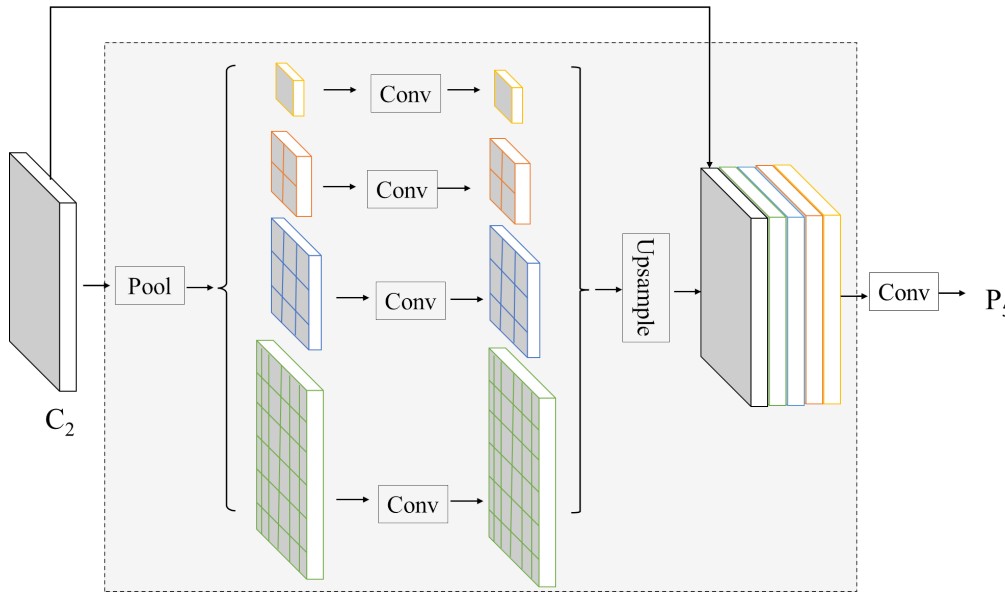

**Figure 9.** Pyramid pooling module.

For the final output of Swin Transformer feature extractor and PPM, the four scales of feature representation are fused on the basis of the FPN, which considers the high spatial resolution detail information and low spatial resolution high-level semantic information. Figure 8 shows that FPN uses topdown fusion and lateral connection to fuse low-resolution feature maps with strong semantic information into high-resolution feature maps with weak semantic information. The input of FPN is $\{C_2, C_3, C_4, C_5\}$, and the output is $\{P_2, P_3, P_4, P_5\}$, and the downsampling multiplier of the four layers is still $\{4, 8, 16, 32\}$. Next, the feature maps $\{P_3, P_4, P_5\}$ are upsampled to the size of the $P_2$ feature map, and the fusion operation is performed to obtain the final fused feature map. The Softmax operation is performed on the fused feature map to obtain the score map of the building.

### 2.2.4. Loss Function

Once the teacher network model is acquired, its parameters are no longer updated due to the limitation of $D_n$ dataset availability. The parameters of the student network are updated using the $D_c$ dataset and the teacher network guide during the training of the student network. The loss function in this work comprises two components, which are teacher–student knowledge distillation and semantic segmentation loss.

Knowledge distillation loss was used to induce knowledge transfer between the teacher and student networks, enhancing the generalization capability of the student network considering building representation. Specifically, knowledge distillation loss can be effectively used to close the distance between the student and teacher network feature maps in the feature extraction process of buildings. The knowledge distillation loss is denoted in this study as:

$$L_{\mathrm{d}} = \sum_{j=2}^{4} \sum_{i}^{N} \text{smooth } L_1\left(\mathrm{P}_j^t - \mathrm{P}_j^s\right), \tag{9}$$

$$\text{smooth } L_1, (x) = \begin{cases} 0.5x^2, & \text{if } |x| < 1, \\ |x| - 0.5, & \text{otherwise. ,} \end{cases} \tag{10}$$

where distillation loss is performed between the multi-scale feature maps of the teacher network and the student network FPN outputs. $\mathrm{P}_j^t$ denotes the jth layer feature map of the teacher network feature pyramid, $\mathrm{P}_j^s$ denotes the jth layer feature map of the student network feature pyramid, and the difference between the $\mathrm{P}_j^t$ and $\mathrm{P}_j^s$ feature maps is measured using the smooth L1 loss metric.

The building extraction task is as a two-class image segmentation task. Thus, the binary cross-entropy (BCE) loss can be used as the semantic segmentation loss in this paper, which can be expressed as:

$$l_s = -\frac{1}{N} \sum_i^N (g_i \log p_i + (1 - g_i) \log(1 - p_i)), \tag{11}$$

where $p_i$ is the predicted probability of the $i$th position in the image, $g_i$ is the truth value of the $i$th position, and $N$ is the overall number of pixels in the image.

Ultimately, the overall loss used to maintain knowledge extraction capabilities and building segmentation capabilities is

$$L_{\text{total}} = L_{\text{d}} + L_{\text{s}}. \tag{12}$$

## 3. Results

### 3.1. Experimental Settings and Evaluation Metrics

The trained model predicts building roofs with a bias due to the effect of sample noise. Noise correction was conducted for high-rise buildings in some areas of Wenzhou City Center to reduce the impact of noise on the results. Training the model directly on the corrected data solves the problem of high-rise building offset but does not identify previous building types. Thus, the multiscale distillation approach is proposed in this paper to address the problem of high-rise building offsets while retaining the generalization capability of the original model.

To verify the effectiveness of the proposed method in solving this problem, the semantic segmentation model on a corrected dataset and the distillation method for semantic segmentation are first compared. Experiments are conducted to verify that the proposed method can solve the problem of building offsets and migrate the generalization capability of the teacher network. Among the semantic segmentation methods with advanced performance that were compared were OCRNet [36], FPN [37], PSPNet [38], UNet [39], DeepLabV3+ [40], and UPerNet [34]. The methods compared in the experiments combining prior knowledge were UPerNet (baseline) [34], teacher networks, networks based on fine-tuning of teacher networks, DML [41], and CWD [42]. The methods that combine knowledge all use UPerNet with Swin-transformer as the feature extractor as a baseline to ensure the feature extraction performance of the network and thus to fairly validate the methods. Second, the results of the different methods are visualized and qualitatively analyzed. Finally, an ablation experiment was conducted to evaluate the contribution of the proposed method to tasks.

Four metrics, namely IoU, F1sorce, Precision, and Recall, were used to verify the performance of the different models in building extraction. For the binary classification task of building extraction, each pixel belongs to one of the categories. The confusion matrix formed after the calculation between the predicted results and the labels contains the following four values: True Positive, False Negative, False Positive, and True Negative. The calculation of the four metrics can be represented by the following. Naturally, large values of the metric indicate improved model performance:

$$\text{Precision} = \frac{\text{True Positive}}{\text{True Positive} + \text{False Positive}}, \tag{13}$$

$$\text{True Positive Recall} = \frac{\text{True Positive}}{\text{True Positive} + \text{False Negative}}, \tag{14}$$

$$\text{F1score} = 2 \times \frac{\text{Precision Recall}}{\text{Precision} + \text{Recall}}, \tag{15}$$

$$\text{IoU} = \frac{\text{True Positive}}{\text{False Positive} + \text{True Positive} + \text{False Negative}}. \tag{16}$$

*3.2. Experimental Parameter Settings*

The GPU platform where the experiments were computed was the NVIDIA RTX3090. In the experiments, all methods resize the input data to a size of $512 \times 512$ and set the batch size to 4, and the number of training iterations is 60,000. All methods use random flip enhancement during training. The AdamW [43] optimizer was used for training to facilitate the easy convergence of the method using the Swin Transformer [25] feature extractor, where AdamW has an initial learning rate of $6 \times 10^{-5}$, $\beta_1 = 0.9$, $\beta_2 = 0.999$ and a weight decay coefficient of 0.01. The strategy of learning rate decay applies to the Poly strategy.

*3.3. Experimental Results*

3.3.1. Quantitative Analysis

Experiments comparing the proposed method with semantic segmentation methods are first conducted, and the results are shown in Table 1. The results of the experiments show that the proposed method achieves the best performance in all four evaluation metrics compared with other advanced semantic segmentation methods. Concretely, the proposed method improves 4.39% in IoU, 2.72% in F1 score, and 2.31% and 3.33% in Precision and Recall, respectively, compared to the UPerNet method using the same feature extractor (Swin Transformer). When compared with the advanced performance OCRNet with HRNet-48 [44] as the feature extractor, the proposed method has 5.25% and 3.27% higher IoU and F1 scores, respectively, and 2.60% and 4.13% improvement in Precision and Recall, respectively. Compared to the state-of-the-art semantic segmentation methods (SegFormer [26], STDC [45], ConvNeXt [46]), our proposed method is 3.91% and 2.42% higher in IoU and F1 score metrics, respectively, when compared to the ConvNeXt method, which has the highest performance among the state-of-the-art methods. The semantic segmentation method used for comparison only trains on noise-corrected data; thus, this method does not obtain the generalized knowledge of the previous model. The test data used in this study are spatially discontinuous with the training data. Thus, the semantic segmentation method for comparison leads to limited model performance due to the reduced generalization capability of the model. The proposed method can gain knowledge from the teacher network through distillation in favor of building extraction, thus facilitating improving the performance of the proposed method on test data. Based on the above analysis, the proposed method is more suitable for the task application, fits the extraction of buildings throughout the city of Wenzhou, and has better generalization performance relative to other methods.

**Table 1.** Comparison of experimental results for the proposed method with semantic segmentation methods.

| Method | Backbone | Precision | Recall | F1 Score | IoU |
|--------|----------|-----------|--------|----------|-----|
| UNet [39] | - | 85.12 | 78.92 | 81.9 | 69.35 |
| PSPNet [38] | ResNet-101 [47] | 84.87 | 84.64 | 84.76 | 73.55 |
| FPN [37] | ResNet-101 [47] | 84.76 | 82.02 | 83.37 | 71.48 |
| UPerNet [34] | ResNet-101 [47] | 85.28 | 84.95 | 85.11 | 74.08 |
| DeepLabV3+ [40] | ResNet-101 [47] | 84.97 | 84.05 | 84.51 | 73.17 |
| OCRNet [36] | HRNet-48 [44] | 86.62 | 86.58 | 86.6 | 76.36 |
| UPerNet [34] | Swin Transformer [25] | 86.91 | 87.38 | 87.15 | 77.22 |
| SegFormer [26] | MIT-B5 | 83.57 | 80.19 | 81.84 | 69.26 |
| STDC [45] | STDC2 | 83.13 | 84.38 | 83.75 | 72.05 |
| ConvNeXt [46] | ConvNeXt-L | 88.35 | 86.57 | 87.45 | 77.70 |
| Proposed | Swin Transformer | 89.22 | 90.71 | 89.87 | 81.61 |

Another set of comparative experiments comparing different methods of teacher network knowledge combination are shown in Table 2. First, the experimental results in the table show that all four metrics verified by the proposed method reached the highest. The

proposed method showed a 4.39% increase in IoU and a 2.72% increase in F1 score compared with the baseline network. In comparison with the results of the teacher-based fine-tuned network, the proposed method has a 2.83% high IoU and 1.74% high F1 score. Second, other approaches to the distillation for semantic segmentation (DML [41] and CWD [42]) not only show no increase in performance but also have a substantially large performance gap with the proposed approach. These results indicated that the proposed method achieves higher performance and has the same advantages over comparable methods.

**Table 2.** Comparison of experimental results for the proposed approach with the combined teacher knowledge approach.

| Method | Backbone | Precision | Recall | F1 Score | IoU |
|---|---|---|---|---|---|
| UPerNet [34] | Swin Transformer [25] | 86.91 | 87.38 | 87.15 | 77.22 |
| TEACHER | Swin Transformer [25] | 88.37 | 86.86 | 87.6 | 77.84 |
| Fine-tune | Swin Transformer [25] | 89.06 | 87.07 | 88.13 | 78.78 |
| DML [41] | Swin Transformer [25] | 87.54 | 86.34 | 86.93 | 76.89 |
| CWD [42] | Swin Transformer [25] | 87.36 | 88.09 | 87.72 | 78.13 |
| Proposed | Swin Transformer [25] | 89.22 (+2.31) | 90.71 (+3.33) | 89.87 (+2.72) | 81.61 (+4.39) |

Further analysis was conducted on the basis of the above results. First, UPerNet is the result obtained by direct training based on the adjusted data. However, the number of noise corrected samples is limited; thus, effectively generalizing the model obtained from the training of samples from the downtown area to the entire city of Wenzhou is difficult. The teacher network TEACHER was trained on a random sample drawn from the entire city of Wenzhou, which contained a full range of building types but included a considerable amount of noise. Therefore, the performance of UPerNet and TEACHER has limitations due to certain drawbacks of both samples. Second, the DML [41] and CWD [42] methods, which combine knowledge from the TEACHER network, also capture knowledge from the TEACHER through distillation. However, DML [41] and CWD [42] are direct ways of keeping the student network close to the distribution probabilities of the teacher network. This knowledge of probability distributions is hard and can prompt the training of student networks. However, teacher networks can also contain errors; thus, hard knowledge simultaneously creates interference in the training of student networks. Finally, the proposed method aims to distill the $P_2$, $P_3$, and $P_4$ feature layers of the FPN. On the one hand, distillation in the key feature layer rather than in the probability layer at the end of the network softens the distilled knowledge. On the other hand, the use of a relation-based distillation loss design ensures effective distillation of knowledge. Overall, the proposed approach effectively combines the advantages of the current data set with the generalization capabilities of the TEACHER model.

### 3.3.2. Qualitative Analysis

The results of some of the compared methods are demonstrated (as shown in Figure 10) to enable a visual comparison. The figure reveals that the proposed method of building extraction is more accurate and misses less compared with other methods. First, compared with the semantic segmentation method, OCRNet is lacking in the third rows of results and incorrectly identifies the background as a building in the first row of results. Second, the results are relatively more complete compared with the distillation method CWD. Third, as can be seen from Figure 10, the model trained in the samples containing noise is significantly biased in the identification of buildings and does not accurately predict the roofs of buildings. An offset in the recognition results of the high-rise buildings is also observed (the segmentation results include the sides of the buildings).

The above comparison and Figure 10 show that the proposed method extracts buildings with a high accuracy and without the problem of high-rise buildings offset. This result occurs due to two reasons. First, training on a noise-corrected dataset can reduce the

noise impact of high-rise building bias. Meanwhile, if trained only in combination with the noise-corrected dataset, then the model failed to identify certain types of buildings and demonstrated poor generalization. Second, the teacher network is rich in the types of buildings used for training despite the presence of noise. Therefore, the proposed method combines the advantages of both by distillation, solving the problem of prediction offset of high-rise buildings and having superior generalization capability. The results demonstrate the capability of the proposed approach to distill the generalized knowledge of the teacher model and combine the strengths of the current data simultaneously. This approach results in improved overall building identification performance.

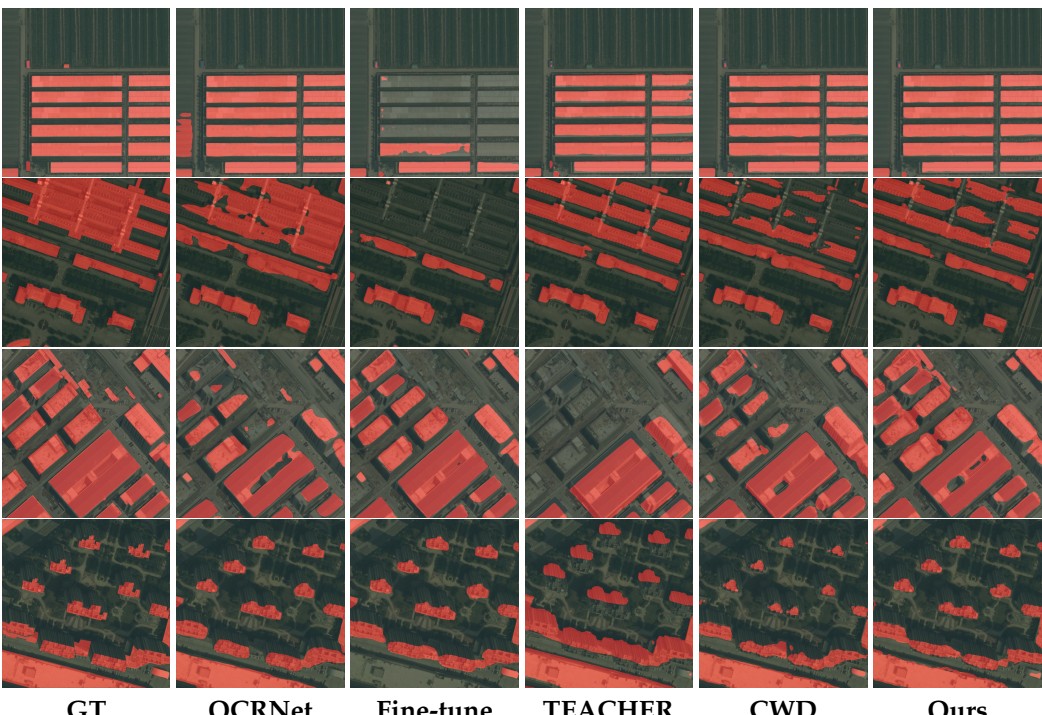

**Figure 10.** Visualization of the superimposed images of the results of the different building segmentation methods. One column represents a method, and the first column represents ground truth. Buildings are shown in red.

## 4. Discussion

The effect of multiscale knowledge distillation on the results of the method when distillation is conducted at different feature layers is discussed in this section. Feature layers are used to perform multiscale knowledge distillation; thus, a combination of feature layers at different scales is used to verify that the best combination of $P_2$, $P_3$, and $P_4$ is used. Regarding the applicability of the proposed method on other models, the applicability of the method on other methods is verified by performing multiscale knowledge distillation based on different semantic segmentation methods. The data and settings of the experiments are consistent with Sections 3.1 and 3.2, respectively.

### 4.1. Experiment of Different Features for Distillation

Overall ablation experiments are performed on the proposed method. Therefore, the effects on the experimental results considering the use of distillation and multiscale distillation are discussed, and the experimental results are shown in Table 3. The experimental results reveal that the use of only a single scale of distillation and the addition of multiple scales improved the accuracy of the final results. The most significant enhancement was achieved using distillation, with a 3.09% increase in IoU, compared with the baseline. Meanwhile, adding multiscale for distillation resulted in a 1.3% increase in IoU compared with using but-scale distillation. This finding indicates that knowledge distillation from

the teacher network markedly improves the performance of the proposed method, and the performance-enhancing knowledge is implicit in the multiscale features. Thus, the use of multiscale features and distillation play a significant role in the proposed method.

**Table 3.** Experimental results of overall ablation study.

| Methods | Precision | Recall | F1 Score | IoU |
|---|---|---|---|---|
| Baseline | 86.91 | 87.38 | 87.15 | 77.22 |
| Proposed (+distill) | 88.75 | 89.4 | 89.08 | 80.31 |
| Proposed (+MS + distill) | 89.22 | 90.71 | 89.87 | 81.61 |

The previous analysis reveals that multiscale features provide valid knowledge for the proposed approach. Experiments were conducted on different combinations of scales (the experimental results are shown in Table 4) to understand the impact of multiscale features on the proposed approach. The experimental results show that the $P_5$ feature distillation effect is the lowest boost when using a single scale feature. Simultaneously, the inclusion of the $P_5$ scale introduces minimal performance gains in the distillation of knowledge at multiple scales that include the $P_5$ feature. Moreover, in multiscale distillation without the inclusion of $P_5$ feature, the model performance gradually increases with the number of scales. This increase is due to the presence of $P_5$ features that are at the top level and cat $P_2$, $P_3$, and $P_4$ in the structure of the UPerNet. Therefore, feature distillation in $P_5$ does not improve the performance of the subsequent segmentation significantly. This finding indicates that multiscale features are effective for knowledge distillation, specifically using $P_2$, $P_3$, and $P_4$.

**Table 4.** Distillation using different combinations of features.

| $P_2$ | $P_3$ | $P_4$ | $P_5$ | Precision | Recall | F1 Score | IoU |
|---|---|---|---|---|---|---|---|
| $\checkmark$ | | | | 88.75 | 89.4 | 89.08 | 80.31 |
| | $\checkmark$ | | | 88.56 | 89.56 | 89.06 | 80.28 |
| | | $\checkmark$ | | 88.78 | 89.57 | 89.17 | 80.36 |
| | | | $\checkmark$ | 87.95 | 88.96 | 88.45 | 79.29 |
| $\checkmark$ | $\checkmark$ | | | 88.8 | 89.71 | 89.25 | 80.59 |
| $\checkmark$ | | $\checkmark$ | | 88.8 | 89.47 | 89.13 | 80.4 |
| | $\checkmark$ | $\checkmark$ | | 88.81 | 89.61 | 89.21 | 80.52 |
| $\checkmark$ | $\checkmark$ | $\checkmark$ | | 89.22 | 90.71 | 89.87 | 81.61 |
| $\checkmark$ | $\checkmark$ | | | 88.64 | 89.82 | 89.23 | 80.55 |
| $\checkmark$ | | $\checkmark$ | $\checkmark$ | 88.33 | 88.5 | 88.41 | 79.23 |
| $\checkmark$ | $\checkmark$ | $\checkmark$ | $\checkmark$ | 89.38 | 89.18 | 89.28 | 80.64 |

*4.2. Method Applicability and Model Complexity*

Multiscale knowledge distillation for UPerNet with ResNet-101 and FPN with Swin Transformer is performed to verify that the proposed distillation would also work for other methods, and the results are shown in Table 5. The table reveals that, although UPerNet + MS + distill has a different feature extractor than the teacher network, the proposed method is equally effective in improving the performance of the student network, with significant improvements in all four validation metrics. Furthermore, the IoU of the proposed method is improved by 3.29% compared with FPN when the same feature extractor and different decoders are used (FPN+ MS + distill). Overall, the proposed method is not only adaptable to different backbones but also equally effective using different decoders. Thus, the proposed method can be adapted to different methods and demonstrates transferability.

**Table 5.** Results of multiscale distillation applied to different methods.

| Methods | Backbone | Precision | Recall | F1 Score | IoU |
|---|---|---|---|---|---|
| UPerNet | ResNet-101 | 85.28 | 84.95 | 85.11 | 74.08 |
| UPerNet + MS + distill | ResNet-101 | 86.68 (+1.40) | 86.99 (+2.04) | 86.84 (+1.73) | 76.74 (+2.66) |
| FPN | Swin Transformer | 87.54 | 86.85 | 87.19 | 77.3 |
| FPN + MS + distill | Swin Transformer | 88.96 (+1.42) | 89.54 (+2.69) | 89.25 (+2.06) | 80.95 (+3.29) |

Finally, the different methods of FLOPs and parameter size are discussed, and the results are shown in Table 6. The table reveals that the UPerNet + Swin Transformer method is more complex to run and has a larger parameter size than the other methods. The proposed method uses the same structure as UPerNet + Swin Transformer for the student network and does not add any other modules; therefore, the parameter size and FLOPs are the same as in the baseline. In addition, the feature extractor of Swin Transformer was finally used in the process of building recognition considering building extraction performance in practice.

**Table 6.** Parameter size and floating point operations (FLOPs) for different methods.

| Methods | Backbone | FLOPs | Params |
|---|---|---|---|
| UPerNet [34] | Swin Transformer [25] | 298.56 G | 121.3 M |
| OCRNet [36] | HRNet-48 [44] | 161.75 G | 70.35 M |
| FPN [37] | Swin Transformer [25] | 172.13 G | 103.8 M |
| Proposed | Swin Transformer [25] | 298.56 G | 121.3 M |

## 5. Conclusions

A multiscale feature knowledge distillation method is proposed in this paper. This method distills semantic knowledge regarding buildings from trained teacher networks containing noisy labels to reduce the impact of noise-containing building datasets on semantic segmentation methods. The most direct way to reduce the impact of noise is to adjust the samples containing noise. However, a large amount of noise correction introduces a large amount of work, and ensuring the generalization of the model for building extraction across the Wenzhou region is difficult due to the adjustment of small areas. Therefore, multiscale distillation is performed from the teacher network to transfer the generalization performance of the teacher network to maintain effective generalization performance for building extraction of the entire Wenzhou region. Meanwhile, the student network acquires the capability to resist noise on samples corrected for noise. Through experimental evaluation, the proposed method shows better performance considering evaluation metrics and is more applicable compared with semantic segmentation and knowledge distillation methods. In addition, the experiments show the effectiveness of the method and demonstrate that the proposed method can be applied to other methods and introduce enhancements to other semantic segmentation methods.

In future work, the proposed method will be explored in the following ways. First, samples containing noise as a form of the weak label are treated and investigated in a self-supervised or semi-supervised way. Second, knowledge distillation in combination with suitable multimodal data will further improve model performance.

**Author Contributions:** Conceptualization, G.X. and J.C.; methodology, G.X. and G.S.; software, Y.G.; validation, G.X. and M.D.; formal analysis, Y.G.; investigation, M.D.; resources, G.S.; data curation, G.S.; writing—original draft preparation, G.X.; writing—review and editing, G.X. and J.C.; visualization, Y.G.; supervision, J.C.; project administration, J.C.; funding acquisition, J.C. All authors have read and agreed to the published version of the manuscript.

**Funding:** The work was supported by the National Key Research and Development Program of China, Grant No. 2020YFA0713503 and the National Natural Science Foundation of China, Grant No. 42071427.

**Data Availability Statement:** Not applicable.

**Conflicts of Interest:** All authors declare no conflict of interest.

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
