# Peer review of "Improving Building Extraction by Using Knowledge Distillation to Reduce the Impact of Label Noise"

_remotesensing, doi:10.3390/rs14225645_

Round 1

Reviewer 1 Report

1.The Dc data set shown in Figure 3, in the first row and third column of the image, verified by Tianditu Map, Baidu Map, Gaode Map and Google Map, the top-right ground objects of the picture are sports fields and swimming pools, but the second row and third column correspond to the label is marked as a building. It means that there maybe wrong labels in the dataset.

2. Lines 181-184, segmentation loss and distillation loss maybe wrongly defined. The segmentation loss is the loss between the student network and the ground truth, and the distillation loss is the loss between the student and the teacher network.

3. The labels in the dataset used for teacher subnetwork training are inaccurate, so can the training of the model fit? Is the model available? If the Teacher subnet is required to provide generalization ability, why not use a pre-trained model with generalization ability trained on ImageNet?

4. The networks in the semantic segmentation method comparison part (FPN, UNet, DeepLab V3+) in Table 1 are old networks, please compare them with the new networks in the past two years.

Reviewer 2 Report

The research is well structured and of high interest.

The reviewer suggests the few improvements:
- Figure 1: contains two arrows to be removed
- Material and Method section: This section should not be focused in the description of the case study. The case study should be reported in the Results section. General aspects of it may be added in the introduction to point towards the research objectives or the problems to be solved, not as a complete description of the case itself. 
- Line 62: there is a reference to Figure 1 that says: "[...] and mislabeling is inevitable when used with current images".  Please, explain to which part of Figure 1, ("a" or "b") this sentence is related to. Also, add a more complete description of Figure 1 in the text body or in the Figure 1 description. Why we see ground labeling in part (a), which is described as "Noisy building label", and roof labeling in part (b), which is described as "Corrected building label"? At this stage it is not clear.
- Line 125: Please describe the source of the used "remote sensing images" . Plese provide information about the instrument used for image acquisition and where we can find them to replicate the experiment. Are they part of a previous project? Are public domain?
-Line 126: "Red, green, and blu bans" is referred to the color model RGB? If so, please write "Red, green, and blu bans (RGB)"
- Line 127-130: Can you please explain what Dn and Dc stand for? Or rewrite as: The large dataset (Dn) with noise and the small pure dataset (Dc). 

- Line 129-130: Can you please provide information about the aereal photos, namely? 1) which instrument was used for the image acquisition. 2) Where we can find the images, if they are public domain?

-Line 130: Please, specify "vector data" source. Where we can find the images, if they are public domain?
- Line 131: who extracted Dc building labels? The authors, someone else did it previously ? 
-Line 133-139: Description not clear. Please, explicitly refer to Dc if at line 134 you are talking about it.
- Figure 4 should appear after the first citation 
- Line 190: replace 2.2.2 with "paragraph 2.2.2" 
- Line 191: same as above
- Line 197: you can provide more info about ViT.
- Figure 8 should appear after been cited in the text
-Table 2 should appear soon after been cited in the text
-Figure 10: Same as above
- From the samples of Figure 10, unfortunately, it is not possible to see if the buildings taken into computation are high-rise building. They look pretty orthogonal to the view point.
